# Are *Bordetella bronchiseptica* Siphoviruses (Genus *Vojvodinavirus*) Appropriate for Phage Therapy—Bacterial Allies or Foes?

**DOI:** 10.3390/v13091732

**Published:** 2021-08-31

**Authors:** Aleksandra Petrovic Fabijan, Verica Aleksic Sabo, Damir Gavric, Zsolt Doffkay, Gábor Rakhely, Petar Knezevic

**Affiliations:** 1Department of Biology and Ecology, Faculty of Sciences, University of Novi Sad, Trg Dositeja Obradovica 3, 21000 Novi Sad, Serbia; verica.aleksic@dbe.uns.ac.rs (V.A.S.); damir.gavric@dbe.uns.ac.rs (D.G.); 2Centre for Infectious Diseases and Microbiology, Westmead Institute for Medical Research, 176 Hawkesbury Road, Westmead, NSW 2145, Australia; 3Department of Biotechnology, University of Szeged, Temesvari krt. 62, H-6726 Szeged, Hungary; zsolt.doffkay@bio.u-szeged.hu (Z.D.); rakhely@brc.hu (G.R.)

**Keywords:** *Bordetella bronchiseptica*, *Vojvodinavirus*, growth inhibition, biofilm, lysogeny

## Abstract

*Bordetella bronchiseptica* is a respiratory animal pathogen that shows growing resistance to commonly used antibiotics, which has necessitated the examination of new antimicrobials, including bacteriophages. In this study, we examined the previously isolated and partially characterized *B. bronchiseptica* siphoviruses of the genus *Vojvodinavirus* (LK3, CN1, CN2, FP1 and MW2) for their ability to inhibit bacterial growth and biofilm, and we examined other therapeutically important properties through genomic analysis and lysogeny experiments. The phages inhibited bacterial growth at a low multiplicity of infection (MOI = 0.001) of up to 85% and at MOI = 1 for >99%. Similarly, depending on the phages and MOIs, biofilm formation inhibition ranged from 65 to 95%. The removal of biofilm by the phages was less efficient but still considerably high (40–75%). Complete genomic sequencing of Bordetella phage LK3 (59,831 bp; G + C 64.01%; 79 ORFs) showed integrase and repressor protein presence, indicating phage potential to lysogenize bacteria. Lysogeny experiments confirmed the presence of phage DNA in bacterial DNA upon infection using PCR, which showed that the LK3 phage forms more or less stable lysogens depending on the bacterial host. Bacterial infection with the LK3 phage enhanced biofilm production, sheep blood hemolysis, flagellar motility, and beta-lactam resistance. The examined phages showed considerable anti-*B. bronchiseptica* activity, but they are inappropriate for therapy because of their temperate nature and lysogenic conversion of the host bacterium.

## 1. Introduction

*Bordetella bronchiseptica* is primarily a respiratory pathogen of domestic, wild, and laboratory animals [1] and is considered to be the evolutionary progenitor of all species of the *Bordetella* genus, including the well-known human pathogens *B. pertussis* and *B. parapertussis* [2]. The prevalence of *B. bronchiseptica* infection is the highest in dogs, cats, swine, rabbits and Guinea pigs, while bordetellosis is rarely documented in horses, monkeys, foxes, rats, opossum, birds or other animals [3,4,5]. Infections caused by *B. bronchiseptica* can manifest from almost asymptomatic to severe purulent pneumonia [6]. Moreover, *B. bronchiseptica* is one of the main causative agents of a few highly contagious and economically significant diseases, including kennel cough, atrophic rhinitis and upper respiratory tract disease (snuffles) [7,8,9]. Although it is generally considered a zoonotic agent, *B. bronchiseptica* rarely causes infection in humans, with the exception of immunocompromised individuals [10].

The vaccine for *B. bronchiseptica*, commonly used since 1970s, is unreliable as it does not provide long-term immunity [11,12]. In addition, the increased degree of resistance to *B. bronchiseptica* isolates against commonly used antibiotics, including β-lactams, macrolides and tetracyclines, has become a serious concern [13,14,15]. In the last few years, the use of bacteriophages in the treatment of bacterial infections has gained the attention of the scientific community [16,17,18,19]. Accordingly, bacteriophage therapy offers an alternative or complementary strategy to conventional antimicrobials for more efficient *B. bronchiseptica* control.

Only several *B. bronchiseptica*-specific phages have been isolated and characterized. A few related *B. bronchispetica* phages from the *Podoviridae* family have been examined in detail, and their genomic analysis indicated involvement in host lysogenic conversion [20]. Recently, phages from the *Myoviridae* and *Podoviridae* families that infect *B. bronchiseptica* were described without showing evidence of lysogeny formation [21,22]. In addition, we reported the first isolation of several phages from the *Siphoviridae* family that are specific to *B. bronchiseptica* [23], but these were not characterized in detail. According to the International Committee for Taxonomy of Viruses (ICTV) report (2019), these phages are classified under the genus *Vojvodinavirus* [24]. They comprise four species—Bordetella virus CN1 (with strains Bordetella phage CN1 and Bordetella phage LK3), Bordetella virus CN2, Bordetella virus FP1 and Bordetella virus MW2. The partial genomic sequence of virus CN1 showed similarity with *Pseudomonas* phages from the genus *Yuavirus* (40%), while transmission electron microscopy, efficacy of plating and RFLP patterns demonstrated high relatedness between examined siphoviruses [23]. Building on this work, we aimed to determine if *B. bronchiseptica*-specific siphoviruses were appropriate for phage therapy. We examined their potential to inhibit bacterial growth and biofilm formation and reduce existing *B. bronchiseptica* biofilm. In addition, we used genomic analysis and conducted lysogeny formation experiments to examine bacterial phenotypic properties after infection with these phages.

## 2. Materials and Methods

### 2.1. Bacterial Strains

For estimation of phage lytic efficacy on biofilm formation and eradication of already formed biofilm, the *B. bronchiseptica* ATCC 10580 strain was used. To examine lysogeny formation, in addition to the reference strain, BbChiot and Bbr3416 animal strains were used [25,26]. All bacteria were stored in Luria Bertani broth (LB) containing glycerol (10% *v*/*v*) at −70 °C. For the experiments, they were inoculated in LB and incubated overnight at 37 °C, if not stated otherwise.

### 2.2. Phages

Five *B. bronchiseptica*-specific siphoviruses (LK3, CN1, CN2, MW2 and FP1) previously isolated from environmental samples and partially characterized, were examined in this study [23]. All phages were propagated, precipitated in 10% PEG6000 and 1 M NaCl and purified by CsCl discontinuous gradient ultracentrifugation (45,000× *g*, 6 h) [27]. The obtained purified stocks were dialyzed and stored at 4 °C for all experiments.

### 2.3. Phage Genome Sequencing and Analysis 

To determine the nature of phages (obligatory lytic or temperate) and other properties, the genomes of five phages of the *Vojvodinavirus* genus (LK3, CN1, CN2, MW2 and FP1) were sequenced and analysed. Their genomic DNA was obtained from the CsCl purified phage suspensions after a standard phenol/chloroform extraction and ethanol precipitation, followed by RNase treatment and DNA reprecipitation. The whole genome sequencing of phages was performed using Illumina technology, while the *de novo* assembly of sequenced fragments was carried out using CLC Genomics Workbench 6.5. ORFs (Open Reading Frames) were predicted using GeneMarkS [28]. The analysis of each ORF was performed using the BLAST algorithm, which enabled the comparison and identification of nucleotide and protein sequences with the sequences deposited in database [29]. The presence of tRNA in the phage genomes was analysed using tRNAscan-SE [30].

### 2.4. Structural Protein Analysis

Phage LK3 virion structural proteins were concentrated from CsCl-purified samples using methanol precipitation [31]. Extracted proteins were treated with Loading Buffer (Tris-HCl, pH 9; 0.5 molL^−1^; SDS 4% *w*/*v*; glycerol 10% *v*/*v*; bromphenol blue 10% *w*/*v*; 2-merkaptoethanol 10% *v*/*v*) prior to heating at 95 °C for 10 min. The separation of proteins and a standard (PierceTM Unstained Protein MW Marker, Thermo Fisher Scientific, Vilnius, Lithuania) were performed using 10% *w*/*v* polyacrylamide gel and Eco-Medium Electrophoresis System (Biometra, Göttingen, Germany) at 200 V for 100 min. SDS-PAGE gel was stained using EZBlue Gel Staining Reagent (Sigma–Aldrich, St. Louis, MO, USA), destained in distilled water and documented by BioDocAnalyze Transiluminator (Biometra, Göttingen, Germany).

### 2.5. Phage Sensitivity to Environmental Factors

The effect of different physical and chemical factors on *B. bronchiseptica* phage (LK3, CN2, MW2 and FP1) infectivity was examined, including temperature, pH, sodium chloride, and urea. In brief, after 30 min of 10^5^ plaque-forming units (PFUs) mL^−1^ phage exposure to different temperatures (4, 37, 45, 55, 65 and 75 °C), pH values (1.5, 3, 5, 7, 9 and 11), urea concentrations (0.5, 1, 3, 5, 7.5 and 10 M) and sodium chloride concentrations (1.5, 3, 5, 7.5 and 10%), treated suspensions were mixed with soft top agar containing the original host *B. bronchiseptica* ATCC 10580. The inoculated top agar was poured onto LB agar and after overnight incubation at 37 °C, the plaques were counted. The experiment was carried out in duplicate on two independent occasions. The results were presented as the percentage of infected phages in the suspensions after treatment. The relation between the starting phage PFUs and the obtained values were presented as mean ± SD.

### 2.6. In Vitro Inhibition of Bacterial Growth and Biofilm Formation by Phages

Three *B. bronchiseptica* siphoviruses (LK3, CN2 and FP1) were selected based on their host range, RFLP profiles and whole genome-sequencing results Their lytic efficacy were examined against the original hosts (*B. bronchiseptica* 10580) according to the method of McLaughlin (2007) [32] modified by Knezevic and Petrovic (2008) [33]. In a sterile microtiter of 96-well flat bottom plates, 50 µL of double strength LB were inoculated with a final bacterial concentration of ~5 × 10^5^ CFU mL^−1^ and an equal volume of 10-fold serial phage dilutions in SM buffer were added (final volume 100 µL). The number of phages in wells were between 1 × 10^2^ and 1 × 10^6^ PFU mL^−1^, and these corresponded to the multiplicity of infection (MOI) in the range of 0.001–10. Controls containing only an inoculated medium with sterile SM buffer (bacterial growth control), or SM buffer and uninoculated double-strength LB (sterility control) were also included. After 24 h incubation at 37 °C, all wells were amended with 0.1% 2,3,5-trifenil tetrazolium chloride (TTC) in a final concentration of 200 µLmL^−1^. After additional 4 h incubation, TTC was reduced to red formazan in wells with viable bacteria. Absorbance was measured by a Multiscan GO (Thermo Fisher Scientific, Lithuania) microtiter plate reader at 540 nm. The absorbances were used to estimate the lytic efficacy of the phages using previously described calculations [33,34]. The experiments were carried out in duplicate in a minimum of two repetitions, and the results were presented as mean values of bacterial growth inhibition (%) with standard deviation.

### 2.7. Phage Effect on B. bronchiseptica Biofilm Formation 

The determination of phage effectiveness on biofilm formation was analysed by a method developed by Knezevic and Petrovic, 2008 [33]. The effect of LK3, CN2 and FP1 phages was examined against the original host, *B. bronchiseptica* ATCC 10580. The content of the well was identical to the one mentioned above in the lytic efficiency experiment. Bacteria were grown in a LB medium, and after incubation every well was washed tree times with PBS (pH 7.4), after which the plates were dried, and the biofilm was fixed for 15 min with 150 μL of methanol. Upon removal of the fixative, the plates were washed with tap water, air-dried, stained for 20 min with 150 μL of 0.4% crystal violet and finally the excessive stain was removed with tap water. The plates were dried at 44 °C for 20 min. The stain was diluted in glacial acetic acid (33%) for 20 min, and the absorbances in the wells were measured using a Multiscan GO microtiter plate reader (Thermo Fisher Scientific, Lithuania) at 595 nm. Every phage–bacteria combination was conducted in at least two repetitions and two independent experiments. The results were presented graphically, and calculations were the same for determining phage lytic efficacy.

### 2.8. Phage Effect on 24-h Old B. bronchiseptica Biofilm

The effect of LK3, CN2 and FP1 phages on already-formed biofilm was examined after bacterial growth in an LB medium at 37 °C for 24 h. After incubation, all wells were washed tree times with PBS (pH 7.4), and double-strength LB, and the same volume of phage 10-fold dilutions were added to the wells. After additional incubation at 37 °C for 24 h, the plates were treated as described for biofilm formation inhibition.

### 2.9. Phage Genome Presence in Bacteria upon Infection

To infect bacteria with phage,* B. bronchiseptica* strains (ATCC 10580, Bbchiot and 3416), which are sensitive to phage LK3, were incubated at 37 °C for 30 min at MOI 0.1, 1, 10 and 100. After incubation, single colonies were picked and cultivated on MacConkey agar and further subcultivated the same way at least twice. To confirm phage infection, genomic DNA were isolated from bacteria using a commercial GeneJET DNA purification kit (Thermo Scientific, Lithuania). Prior to DNA extraction, the cells were washed three times with PBS (NaCl 137.00 mmolL^−1^, KCl 2.70 mmolL^−1^ Na_2_HPO_4_ 4.30 mmolL^−1^, KH_2_PO_4_ 1.47 mmolL^−1^.) to remove potential phages that did not infect bacteria. A PCR was applied for phage genome detection in bacteria, using two pairs of specific LK3 phage primers (Bbr-targeting ribonucleotide–diphosphate reductase subunit alpha and Hem-targeting virion structural protein) and a pair of *B. bronchiseptica* housekeeping gene *recA* primers as a control (Table 1). Thermal cycling conditions were as follows: an initial cycle of 94 °C for 5 min followed by 35 cycles of 94 °C for 30 s, annealing at 55 °C for 30 s, and extension at 72 °C for 30 s, with a final 7 min extension at 72 °C. PCR products were analysed on 1% agarose gel with ethidium bromide. A positive control was phage DNA and a negative one was distilled water. The gels were documented using BioDocAnalyze Transiluminator (Biometra, Germany). In addition to conventional PCR, restriction enzyme digestion was used to confirm phage infection. Briefly, DNA of lysogenic and non-lysogenic bacteria was treated with SalI fast-digest enzyme (Thermo Scientific, Lithuania) for 15 min at 37 °C, followed by fragments separation on 1% of agarose gel. The SalI enzyme cuts bacterial DNA producing great number of small fragments that appear on the gel as a smear. The phage genome does not contain a SalI-specific sequence, so the enzyme does not digest the phage DNA that appears on the gel as the highest clear single band. The DNA of non-lysogenic bacteria treated with SalI and phage LK3 genomic DNA were used as controls.

### 2.10. Phenotypic Changes of lysogenic B. bronchiseptica 

#### 2.10.1. Biofilm Production of Lysogenic Bacteria 

To determine the biofilm formation potential of lysogenic bacteria, the previously described method by Stepanovic et al., 2000 was used [35]. Briefly, both lysogenic (ATCC10580+, Bbchiot+ and Bbr3416+) and non-lysogenic (ATCC10580, Bbchiot and Bbr3416) bacterial strains were grown in LB medium over 48 h. After incubation, every strain was washed tree times with sterile distilled water and the number of bacteria was adjusted to ~2 × 10^3^, 2 × 10^5^ and 2 × 10^7^ CFU mL^−1^. In two separated microtiter plates, 200 μL of bacterial suspension was added, and both plates were placed at 37 °C; one plate was incubated for 24 h while the second one was incubated for 48 h. The LB medium without bacteria was used as a negative control and a *Pseudomonas aeruginosa* PAO1 reference strain was used as a positive control in the experiment. After incubation, all wells, including controls, were washed three times with PBS (pH 7.4) to remove planktonic cells that were not part of the biofilm. Formed biofilm was fixed with 150 μL of methanol for 15 min, after which the methanol was removed and the plates were dried in an incubator at 44 °C for 20 min. Dried plates were stained with 0.4% crystal violet solution for 15 min, after which the stain was removed by washing the plates with water. The plates were then placed back into the incubator at 44 °C for 15–20 min to dry. After incubation, 150 μL of 33% acetic acid was added in each well to dissolve the stain from biofilm. Absorbance was measured on Multiscan GO (Thermo Scientific, Lithuania) plate reader at 595 nm. The experiment was carried out in triplicate and on three independent occasions. The results were presented graphically, as mean value of absorbance with standard deviation.

#### 2.10.2. Hemolytic Activity of Lysogenic Bacteria 

To determine if there were a significant difference in hemolytic activity between non-lysogenic and lysogenic bacterial strains, the method described by Janda and Abbott, 1993 [36] was used. In this experiment, a 1% erythrocyte solution was extracted from blood (sheep, rat and cow) in 0.9% sodium chloride, after which the erythrocyte solution was mixed with bacterial suspensions (~1.5 McFarland). The mixture was incubated at 37 °C for 48 h after which the released hemoglobin was quantified using the microtiter plate reader Multiscan GO (Thermo Fisher, Lithuania) at 540 nm. The 1% SDS solution was used as a positive control, while 0.9% sodium chloride was used as a negative control. The obtained values were analysed in STATISTICA 12.0, using the Student’s t-test to determine if there were a statistically significant difference in the hemolysis of erythrocytes between non-lysogenic and lysogenic strains.

#### 2.10.3. Susceptibility of Lysogenic Bacteria to Antibiotics

The susceptibility of lysogenic bacteria antibiotics was determined by standard disk-diffusion methods (Kirby–Bauer) and according to CLSI (2007). A total of 20 antibiotics from different classes were used in the experiment: beta-lactams (ampicillin (10 µg), amoxicillin (25 µg), amoxicillin/clavulanic acid (20/10 µg), cefalexin (30 µg), cefoxitin (30 µg), ceftazidime (30 µg), cefotaxime (30 µg) and imipenem (10 µg)); aminoglycosides (amikacin (30 µg), gentamicin (10 µg), streptomycin (10 µg), kanamycin (30 µg), tobramycin (10 µg), netilmicin (30 µg) and neomycin (30 µg)], tetracycline [tetracycline (30 µg) and doxycycline (30 µg)], phenicol [chloramphenicol (30 µg)); fluoroquinolones (ciprofloxacin (5 µg)); macrolides (aztreonam (30 µg)); and sulphonamides (sulfamethoxazole/trimethoprim (23.75/1.25 µg)). Testing was done in two independent experiments and the values (i.e., diameters of inhibition zone by antibiotic) were analysed in STATISTICA 12.0 software using the Student’s *t*-test to determine significant differences in sensitivities to antibiotics between lysogenic and non-lysogenic *B. bronchiseptica* strains. *Escherichia coli* ATCC 25922 was used as a control. Results were expressed the average values of two experiments + SD.

#### 2.10.4. Swimming and Twitching Motility of Lysogenic Bacteria

Flagellar motility, also known as swimming motility, was examined in lysogenic bacteria using the method described by Kerns (2010) [37]. Briefly, bacteria were inoculated in the center of a semisolid LB Agar (0.3%) using a sterile needle. Both plates with lysogenic and non-lysogenic bacteria were incubated at 37 °C for 48 h. Following incubation, the diameter of the colony was measured, and the results presented as average + SE. The flagella-independent form of motility, also known as twitching motility, was examined in lysogenic bacteria using standard LB Agar. Lysogenic and non-lysogenic bacteria were inoculated by a sterile needle in the center of the plate. Following incubation (37 °C for 48 h), the diameter of colony was measured in a Petri dish and medium interface and the results were presented as average + SE. Both swimming and twitching motility experiments were performed in duplicate in two independent occasions and the *Pseudomonas aeruginosa* reference strain PAO1 was used as a control.

#### 2.10.5. Fimbria Production

To better understand the phage role in *B. bronchiseptica* pathogenicity, the production of fimbria in lysogenic bacteria was also examined. For this purpose, the adapted method described by Jain and Chen (2007) was used [38]. Briefly, bacteria were cultured in LB Agar containing 0.5% sodium chloride and subsequently incubated for 5 days at 26 and 37 °C. After incubation, bacteria were suspended in 0.9% sodium chloride to ~1 × 10^9^ CFU mL^−1^. Bacterial suspension was then centrifuged (16,000 × *g*, 10 min) to obtain the pellet. A solution of 0.002% Congo red was used to resuspend the bacterial pellet and the absorbance was measured at 500 nm. The results were presented as a percentage of reduced Congo red that was bound by fimbriae in comparison to the sterile solution without bacteria. *Escherichia coli* ATCC 25922 was used as a control.

## 3. Results

### 3.1. Bordetella Phage LK3 Genome Properties

Complete genomic sequences of phages were deposited with GenBank under accession numbers KX961385 (LK3), KY000218 (MW2), KY000219 (CN2), KY000220 (FP1) and KY000221 (CN1). The genome of LK3 phage consisted of double-stranded DNA with a length of 59,831 bp and G + C composition of 64.01%, with no tRNA detected. Among the total of 78 ORFs (GeneMark), 32 (40.5%) were identified as hypothetical proteins (Figure 1).

The LK3 genome also contains eight ORFs that shows no similarity with sequences from the GenBank database. The predicted ORFs were roughly classified into three functional regions: (i) nucleotide metabolism and DNA replication, (ii) host interaction, and (iii) virion structure and packaging with lysis cassette (Figure 2). Putative genes encoding repressor protein (ORF52) and integrase (ORF53) were detected in the genome, similar to the integrase of Pseudomonas phage MP1412 at 99% and Pseudomonas phage PAE1 at 100%, respectively, which indicated the temperate nature of the phage. Nevertheless, LK3 also possessed genes coding for DNA polymerase, primase and helicase, which are involved in replication. A few genes involved in the host interaction were also detected, such as diguanylate cyclase with GGDEF domain and an anti-restriction enzyme. The lytic cassette of LK3 phage displayed similarity with other phages that infect Gram-negative bacteria and consists of four overlapping genes: spanins, endolysin and holine. Analysis of endolysin sequence revealed the lytic transglycosylase muralytic activity, globular structure, and presence of signal peptide sequence.

The whole genome phylogeny showed high sequence similarity among the five *B. bronchiseptica* siphoviruses. (>88%) and relatively high homology to Pseudomonas Yua-like siphoviruses (>40%); highest similarity was recorded with LKO4 (52.6%) and Yua (44.7%) phages [24]. Based on these results, phages were selected for further characterisation which included testing of sensitivity to different environmental factors (LK3, CN2, FP1 and MW2) and their lytic efficacy against their original host (LK3, CN2, FP1).

### 3.2. Proteomic Characterization of LK3 Phage

Structural proteins of LK3 phage were analyzed by 1D sodium dodecyl sulfate-polyacrylamide gel electrophoresis (SDS-PAGE) and at least 9 protein bands of ~18–100 kDa were detected, including major capsid proteins with a molecular weight of ~32.5  and 54.2 kDa (Figure 2). All bands corresponded to predicted structural proteins in silico, with an acceptable error of molecular weight of less than 5% [39].

### 3.3. Phage Resistance to Environmental Factors

All phages were stable in a pH range of 3–11 for 30 min, and all lost infectivity at pH 1.5 (Figure 3A). Only phage FP1 was affected by 5% NaCl as this concentration reduced the number of virions by more than 50%, while the other phages were resistant even to 10% NaCl for 30 min (Figure 3B). The number of FP1 and MW2 virions was reduced by half at 65 °C for 30 min, while all phages were inactivated at 75 °C (Figure 3C). Similarly, 7.5 M of urea reduced the number of FP1 and MW2 phages by half, while for phages CN2 and LK3 the same effect was obtained with 10 M of urea for 30 min (Figure 3D).

### 3.4. Phage Inhibition of Bacterial Growth and Biofilm Formation

The results of bacterial growth, biofilm inhibition and biofilm removal by LK3, CN2 and FP1 phages are shown in Figure 4.

All bacteriophages exhibited relatively high efficacy of lysis on original host *B. bronchiseptica* ATCC 10580. Even at the lowest MOI, inhibition of bacterial growth was considerable (85%), with maximal inhibition reached at MOI 1 (>99%) (Figure 4A). The results of biofilm inhibition (Figure 4B) and biofilm removal (Figure 4C) indicated that all phages had a similar effect on *B. bronchiseptica* ATCC 10580 biofilm. At the lowest MOI, inhibition of biofilm formation was 70%, while maximal inhibition was reached at MOI 1 and MOI 0.1 (>90%), in LK3/FP1 phages and the CN1 phage, respectively. However, phages showed lower efficacy on 24 h-old biofilm compared to biofilm in the formation stage. The lowest efficacy in biofilm reduction was at MOI 1 for LK3 (approx. 40%). Regardless of the results, the efficacy of phages on existing biofilm was still considerably high, with the highest efficacy reached by the same phage at MOI 0.01 (<80%).

### 3.5. Confirmation of LK3 Infection and Stability of Lysogens

To confirm *B. bronchiseptica* infection with LK3 phage (ATCC 10580+, Bbchiot+ and Bbr3416+), PCR was used for two pairs of phage-specific primers (Bbr and Hem). The expected products, 241 bp for Bbr and 170 bp for Hem primer pairs, were detected in the genomic DNA of LK3 lysogenic bacterial strains (Figure 5A–C). Restriction digestion with enzyme SalI additionally confirmed infection of *B. bronchispetica* strains with LK3 phage (Figure 5D), as the enzyme cut bacterial DNA on a huge number of fragments, but the phage DNA stayed intact and appeared on the gel as the highest uncut fragment. The results of the experiments for stable phage genome persistence in bacteria indicated that *B. bronchiseptica* LK3 phages formed unstable lysogens with the original host ATCC 10580 (Figure 5E). Namely, after the third subcultivation of the lysogenic strain, the expected Hem PCR product was not detected, indicating absence of phage DNA in the strain. The other two lysogenic strains, Bbchiot+ and Bbr3416+, showed greater stability, as the PCR product was detected even after the sixth subcultivation, both in genomic and plasmid DNA.

### 3.6. Lysogenic Conversion of B. bronchiseptica by LK3 Siphovirus 

The potential of both non-lysogenic and lysogenic cells to form biofilm was examined using various starting bacterial numbers (~2 *×* 10^3^, 2 *×* 10^5^ and 2 *×* 10^7^ CFU mL^−1^) (Figure 6A–C). Lysogenic strains showed increased adherent and aggregative growth in broth culture (Figure 6C tube), with an exception for the LK3 lysogenic strain ATCC10580+ and Bbchiot+ at inoculum ~2 *×* 10^3^ CFU mL^−1^ after 24 h.

With the exception of these two cases, at all starting bacterial numbers lysogenic strains showed statistically significant higher biofilm production after 24 h in comparison with non-lysogenic strains (*p* < 0.01). A statistically significant difference was also determined between lysogenic and non-lysogenic strains of *B. bronchiseptica* after 48 h incubation (*p* < 0.01) (Figure 5B). It is also interesting to note that some strains (Bbchiot+ and Bbr3416+) formed 2–4 times more biofilm compared to wild types. The results of other phenotypic testing on lysogenic bacteria (ATCC 10580+) are summarized in Table 2. Upon infection of the ATCC 10580 strain with phage LK3 (ATCC 10580+), an increase of sheep blood hemolysis was noticed, while this activity against rat and cattle blood did not change significantly. The phage increased flagellar motility of Bbchiot strain, while phage infection had no effect on twitching motility and fimbria production. A change in antibiotic susceptibility was also observed, most prominently against amoxicillin/clavulanic acid: as an intermediate-sensitive strain, ATCC 10580 became resistant, and the sensitive Bbchiot strain became intermediately sensitive. A similar change from the intermediate-sensitive strain Bbchiot to resistant was confirmed for ceftazidime. Although LK3 changed the inhibition zone diameter around discs of doxycycline and sulfamethoxazole/trimethoprim, the bacteria remained sensitive. Sensitivity to other antibiotics did not change significantly.

## 4. Discussion

*B. bronchiseptica* siphoviruses had previously been isolated and morphologically characterized, and their host range and RFLP pattern had been determined [23]. All phages displayed mutually high similarity with certain variation, especially regarding RFLP pattern and efficacy of plating. In this study their potential applicability as anti-*Bordetella* agents and their usability in phage therapy were examined.

The *in vitro* lytic efficacy of *B. bronchiseptica*-specific siphoviruses showed up to 99% total bacterial growth inhibition. The method applied can be considered as an estimation of both the planktonic and biofilm phage effect [34,40,41]. The biofilm formation inhibition indicated that *B. bronchiseptica* phages are highly efficient at biofilm prevention (up to 93% for CN1 phage), which was in accordance with the phage inhibition of total bacterial growth; that is, the highest inhibition of both total bacterial growth and biofilm formation by siphoviruses were at MOI 1. Since absolute inhibition of neither total bacterial growth nor biofilm formation inhibition was achieved at MOI10, it is most likely that the phages were temperate.

The 24 h-old *B. bronchiseptica* biofilm removal by phages was lower in comparison to biofilm inhibition. The lower percentage of biofilm reduction by siphoviruses can be explained by fact that biofilm is a complex microbial community in which gene expression and receptors for phages are quantitatively and qualitatively different from those in planktonic cells [42]. The importance of pili in the initial phase of *B. bronchiseptica* biofilm formation was highlighted by Irie et al. [43], and since the LK3 phage displayed DNA sequence similarity with Pseudomonas phage Yua, which has pili as receptors [44], it is very likely that LK3 phage also used these surface structures when adhering to the host cell. Thus, in the complex matrix of biofilm, pili receptors are not available to phages as they are in planktonic cells. Furthermore, bacteria in biofilm are less metabolically active, so phage production is generally reduced as temperate phages integrate with the bacterial genome rather than enter the lytic cycle [45]. Regardless, the lower efficacy of phages in biofilm reduction compared to biofilm formation inhibition, Bordetella phage LK3 generally had a satisfactory effect on biofilm prevention and removal. The relatively high reduction of 24 h-old biofilm can be explained by the fact that extracellular DNA (eDNA), as important component of biofilm matrix [46], occurred after 48 h of *B. bronchiseptica* incubation [47], so the biofilm had not completely matured.

Taking into consideration that different factors can affect the efficacy of phage therapy, primarily the multiplicity of infection (MOI) [48], in this study the lytic efficacy of LK3, CN1 and FP1 phages were examined at different MOI (0.001–10) to determine at which MOI value the best antibacterial effect was achieved. It is interesting that highest efficacy was recorded at lower MOI values (0.001–0.1), indicating the potential integration of phage with the bacterial genome [49]. Many authors such as Knezevic et al. [34], suggested that MOI 10 is the most appropriate for a successful antibacterial phages effect because at this MOI each bacterium is infected by at least one phage following the Poisson distribution. However, when temperate phages are examined, the effect of different MOIs on phage lytic activity is different [15,50]. One example is the lambda phage, for which phage integrase activity directly depends on the concentration of cell proteases and MOI value: a low concentration of protease or high MOI values enhance integrase activity, which promotes phage entry into the lysogenic cycle. On the other hand, at a low MOI value, integrase activity was reduced to a minimum and the phage did not integrate with the bacterial genome but entered the lytic cycle [49]. Taking into account that the phage did not show enhanced lytic properties at MOI 10, it seems that they were temperate, i.e., able to enter the lysogenic cycle like a lambda phage. Good biofilm removal efficacy was recently observed for a Yuavirus-related Pseudomonas phage Zc01, which was superior in comparison to two *Podoviridae* and showed efficacy at a very low MOI, approx. 0.01 [51].

To consider phage properties further, the Bordetella phage LK3 genome was sequenced to reveal genes responsible for lysogenization. The LK3 phage displayed high DNA-sequence similarity with a few Yua-related *P. aeruginosas* phages (MP1412, Yua and LK04) [44,52], and possessed genes for integrase and repressor proteins. These findings indicated that LK3 was capable of integrating into the host’s genome, further confirming its temperate nature and, along with the previously confirmed *pac* site [23], indicating possible phage involvement in transduction [53]. The genome of LK3 also has genes coding for DNA polymerase, primase and helicase and others involved in replication. These genes are part of the replisome, a multiprotein complex that dictates the coordinated synthesis of both leading and lagging DNA [54] and increases replication efficacy [55]. The predicted structural genes corresponded to the protein profile obtained by SDS-PAGE.

For genes involved in host interactions, the most interesting one detected in the LK3 phage genome, apart from integrase and repressor protein, was diguanylate-cyclase with a GGDEF domain, which is involved in global secondary messenger synthesis (cGMP) by controlling motility and biofilm formation in bacteria. This enzyme was previously detected in *Pseudomonas*-specific phages from the genus *Yuavirus* and probably contributes to bacteria properties, including those of *B. bronchiseptica*. The LK3 phage does not possess a reverse transcriptase, which had previously been detected in the genome of the temperate *B. bronchiseptica* phage from the family *Podovirirdae* and generates nucleotide diversity in the *mtd* region, responsible for the specificity of receptor molecules [56]. It is worth highlighting the presence of the gene for anti-restriction protein (responsible for the interaction with restriction enzymes and the tail-structural proteins ORF27 and ORF28) that contains the F5/8 (discodin) domain of eukaryotic coagulation factor, which specifically binds to phospholipids [57]. Proteins with this domain are not present in other Yua-related phages and probably play a role in the interaction with bacterial cell membranes. Lysis cassettes of LK3 phage consist of genes encoding pinholin, endolysin and spanin (Rz protein), which is generally characteristic of phages that infect Gram-negative bacteria. An analysis of the LK3 endolysin sequence confirmed the presence of a signal peptide sequence, supposed to have a role in membrane integration, and that the existing pinholin contributed to the lytic process by accelerating it and causing membrane depolarization [58,59]. The lysin can have practical applications in bacterial growth control with further examination or modification.

Infection experiments confirmed that the LK3 phage genome can persist in the sensitive strains (ATCC 10580, Bbchiot and Bbr3416) because the presence of phage DNA in the bacterial genomic DNA was confirmed. The stability of *B. bronchiseptica* LK3 lysogens was tested through continuous subcultivation of lysogenic strains and infection persistence, i.e., phage DNA presence. Taking into account that the Bbchiot and Bbr3416 strains contained phage DNA after the sixth subcultivation, they formed stable lysogens. Unlike them, the reference strain *B. bronchiseptica* ATCC 10580 lysogen did not show an equal persistence of phage in the bacterial cell: phage DNA was not detected in bacteria after the third subcultivation of the lysogenic strain, indicating that the LK3 phage formed unstable lysognes of the reference strain. The Bordetella phage LK3 displayed a high degree of morphological and genetic similarity with the Yua phage, which cannot form lysogeny. Specifically, Ceyssens et al. [44] attempted unsuccessfully to confirm the presence of the Yua phage in PAO1 colonies from blurry lysogenic strain plaques using PCR and RFLP. The absence of lysogens was also observed for the related alpha-proteobacterium phage, φJL001 [60,61]. The authors indicated that the Yua integrase requires specific physiological conditions or an alternative host for stable lysogen establishment [44]. Other authors only predicted the Yua-related phage’s temperate nature based on the presence of integrase and repressor in the genome but did not try to check this property, which pertains to phages PAE1 [62], S218 [63] and Ab18, 19 and 20 [64]. Thus, this is the first confirmation that Yua-related siphoviruses can form lysogeny, which is not desirable from the point of view of the application of phages in therapy. Considering that prophages almost always influence the bacterial phenotype, the results of this study clearly indicated a potential role for the *B. bronchiseptica* LK3 phage in lysogenic conversion [65].

After examining the LK3 lysogenic strains, it was confirmed that sheep blood hemolysis and flagellar motility were elevated, confirming phage involvement in virulence. Similarly, *B. bronchiseptica* infection with phage LK3 significantly reduced sensitivity to beta-lactam antibiotics. More importantly, we showed that siphoviruses encode diguanylate-cyclase with a GGDEF domain, a key initiator of biofilm formation in bacteria, and we demonstrated hyperbiofilm production in lysogenic bacteria. These data clearly indicate the possible role of siphoviruses in the biofilm formation of *B. bronchiseptica*, which is thought to support bacterial survival and persistence in the mammalian respiratory tract. The mechanisms of these changes, however, should be further examined, as well as the effect of these phages to better understand the phage contribution to bacterial virulence. Taking into consideration that *B. bronchiseptica* is known to modulate strongly both innate and adaptive immune responses in mammals [66] and that temperate phages can contribute to the attenuation of basic immune defenses such as phagocytosis [67], future studies should also investigate the potential role of siphoviruses in the immunomodulation and pathogenesis of *B. bronchiseptica* infections [68].

Since the phages are involved in the lysogenic conversion of *B. bronchiseptica*, their susceptibility to environmental factors was also determined. The phages tolerate a range of pH, high temperatures and concentrations of salt and urea, and the demonstrate the ability to persist even in environments unfavorable to the host bacteria. It is well known that among phages, members of family *Siphoviridae* possess the most stable virions [69]. Maintenance of infectivity at high temperatures (65 °C) is of great importance, as we have shown here that these phages are involved in bacterial virulence, and pasteurization is not a barrier to their transmission.

## 5. Conclusions

In summary, this is the first evidence of the antibacterial activity of *B. bronchiseptica* bacteriophages. the first genomic analysis of a *B. bronchiseptica*-specific phage from the family *Siphoviridae* and the first confirmation of lysogeny formation by Pseudomonas phage Yua-related bacteriophages. The phages are not appropriate for phage therapy because of their temperate nature despite the high total bacterial growth inhibition and anti-biofilm effect. They affect bacterial biofilm production, hemolytic properties, motility and antibiotic susceptibility, indicating involvement in lysogenic conversion. Furthermore, the phages are stable under unfavorable conditions for *B. bronchiseptica*, so they can persist in the environment and lysogenize new hosts.

## Figures and Tables

**Figure 1 viruses-13-01732-f001:**
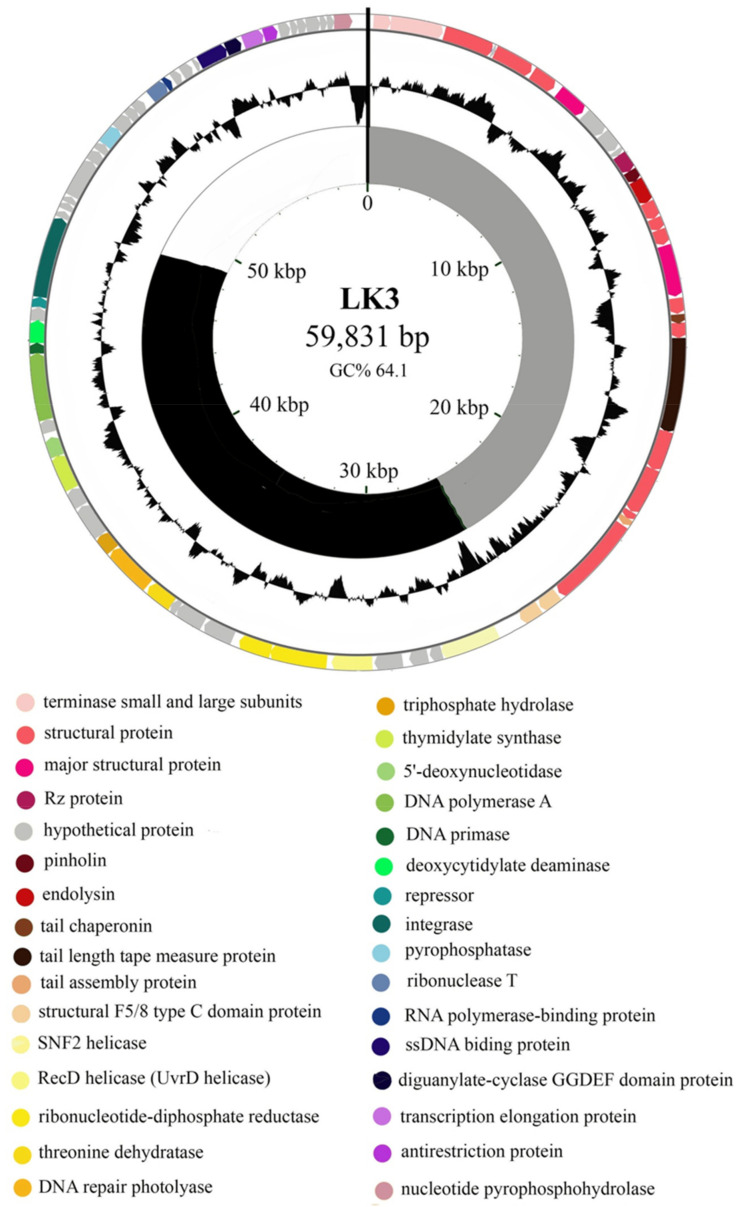
Genomic map of circularly permuted vB_BbrS_LK3 Bordetella phage dsDNA: the outer ring illustrates the encoded genes/ORFs with putative functions indicated in the legend below the circle; the central ring represents GC% content in the genome; the inner ring represents modules: (i) nucleotide metabolism and DNA replication (black); (ii) host interaction module (white), and (iii) particle structure and packaging module with lysis cassette (grey).

**Figure 2 viruses-13-01732-f002:**
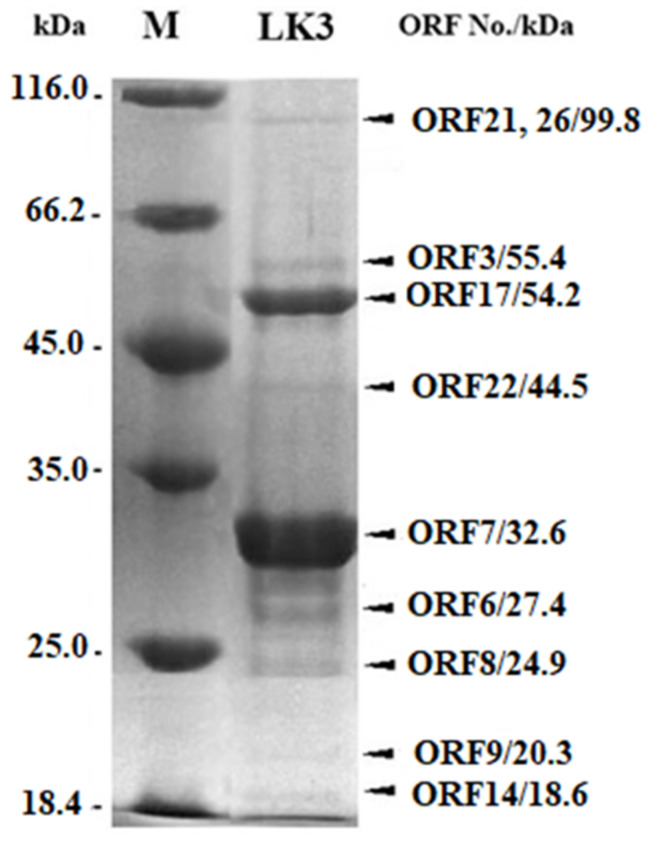
1D SDS-PAGE of phage LK3 virion proteins; M—protein marker. Each band is designated with appropriate ORF and the difference between the predicted and obtained proteins’ MW was less than 5%.

**Figure 3 viruses-13-01732-f003:**
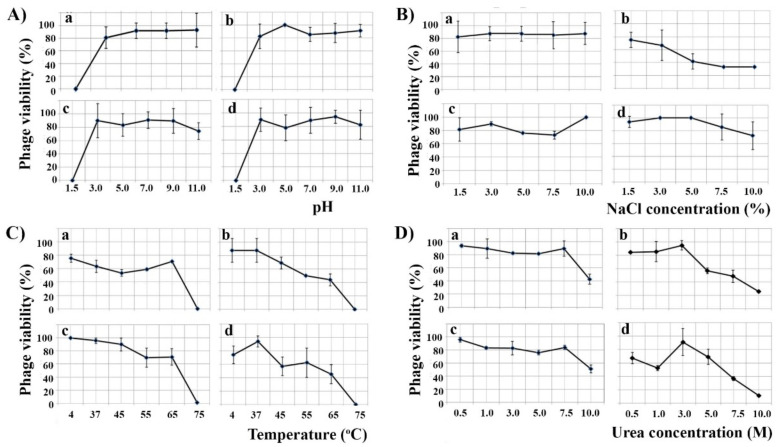
Effect of different pH (**A**), NaCl concentrations (**B**), temperatures (**C**) and urea concentrations (**D**) on the infectivity of *B. bronchiseptica* phages CN2 (**a**), FP1 (**b**), LK3 (**c**) and MW2 (**d**).

**Figure 4 viruses-13-01732-f004:**
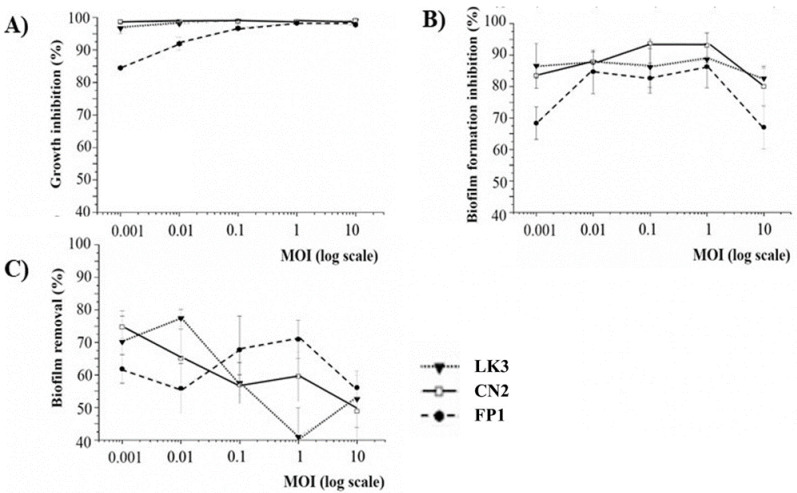
Bacterial growth inhibition (**A**), inhibition of biofilm formation (**B**) and removal of *B. bronchiseptica* ATCC 10580 biofilm (**C**) by siphoviruses: LK3 (▼), CN2 (□) and FP1 (●).

**Figure 5 viruses-13-01732-f005:**
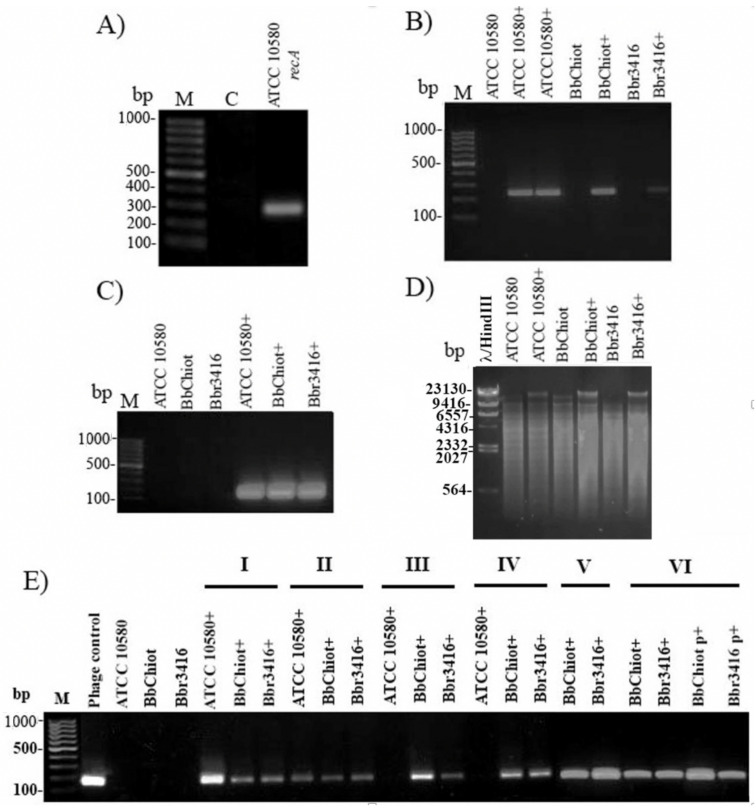
Confirmation of phage DNA presence in genomic DNA of bacterial strains ATCC 10580, BbChiot and Bbr3416 upon infection: (**A**) Quality control of bacterial DNA with Bbr-recA primers; (**B**) Infection confirmation with specific primers Bbr-R and Bbr-F (product size 241 bp); (**C**) Infection confirmation with specific primers Hem-R and Hem-F (product size 170 bp); (**D**) Infection confirmation of bacterial strains ATCC 10580, BbChiot and Bbr3416 with restriction digestion with enzyme SalI; phage DNA in lysogenic strain stayed intact.; (**E**) Presence of viral DNA in bacterial genomic DNA of strains ATCC 10580, BbChiot and Bbr3416 during six successive transfers onto a new medium confirmed by primer pairs Hem (I-VI); M—DNA ladder 100 bp; (**C**) a negative control; (+) lysogenic strain with phage.

**Figure 6 viruses-13-01732-f006:**
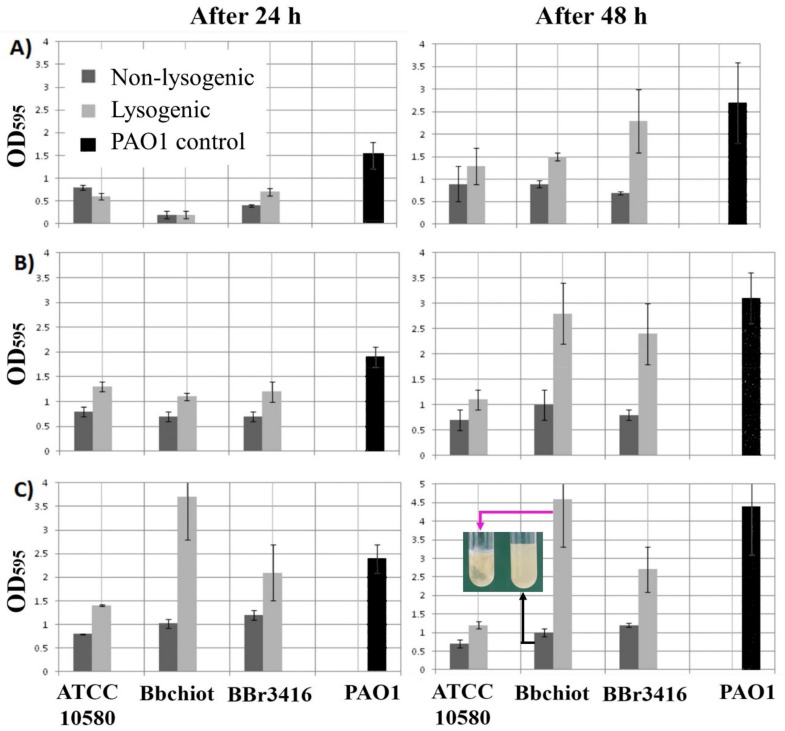
Biofilm production by non-lysogenic and lysogenic *B. bronchiseptica* strains after 24 and 48 h incubation. Starting concentration of bacteria ~2 *×* 10^3^ CFU mL^−1^ (**A**), ~2 *×* 10^5^ (**B**) and 2 *×* 10^7^ (**C**).

**Table 1 viruses-13-01732-t001:** Primers used in the study.

Label of Primer	Sequence (5′→3′)	Expected Product (bp)	Reference
Bbr-F	TGA CTT CAT GGT TGC CGT TC	241	[23]
Bbr-R	TCG GGA GCG TGA TTT CAG TA		
Hem-F	ACG CCC GAA CCG TTA TTT GG	170	This study
Hem-R	CAT TTC CCC GCA ACT CGA CA		
RecA-F	ATG GCG ACA ACG AGG TCG AA	263	[23]
RecA-R	CAG CAG GTC GGT CAG GTT GA		

**Table 2 viruses-13-01732-t002:** Phenotypic changes of bacterial characteristics upon infection with Bordetella phage LK3.

		Strains			
Characteristics		ATCC10580	ATCC10580+	Bbchiot	Bbchiot+
**Hemolysis** **(%)**	Sheep blood	14.3 ± 3.7	26 ± 0.28 *	N.T.	N.T.
	Rat blood	47 ± 6.8	38.1 ± 6.1	N.T.	N.T.
	Cattle blood	66.95 ± 7	59.1 ± 12.02	N.T.	N.T.
**Motility** **(mm)**	Swimming	14.6 ± 0.7	14.4 ± 1.6	66.9 ± 2.5	78.2 ± 1.8
	Twitching	4.0 ± 0.0	4.3 ± 0.2	5.0 ± 0.3	4.1 ± 0.2
**Fimbria production**	Congo-red reduction (%)	10.0 ± 2.1	11,8 ± 2	9.1 ± 1.3	8.5 ± 2.3
**Susceptibility to antibiotics ^1^ (mm)**	Amoxicillin/clavulanic acid	15.8 ± 2.1	12.1 ± 0.9 *	18.8 ± 1.0	15.3 ± 0.9 *
	Ceftazidime	17.3 ± 3.0	16.0 ± 2.0	20.0 ± 1.0	13.5 ± 0.5 *
	Doxycycline	43.0 ± 1.2	38.3 ± 1.0	33.5 ± 1.7	35.9 ± 3.6
	Sulfamethoxazole/Trimethoprim	25.5 ± 4.1	19.6 ± 1.1	0	0

^1^ Antimicrobials that do not cause change in inhibition diameter are not shown in the table; N.T. stands for not tested; Asterisks * indicate significantly different values based on Student’s *t*-test (*p* ≤ 0.05).

## Data Availability

The genomic sequences of the *Bordetella bronchiseptica* viruses are available in Genbank under the accession numbers KX961385 (LK3), KY000218 (MW2), KY000219 (CN2), KY000220 (FP1) and KY000221 (CN1).

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
