# Peer review of "Are *Bordetella bronchiseptica* Siphoviruses (Genus *Vojvodinavirus*) Appropriate for Phage Therapy—Bacterial Allies or Foes?"

_viruses, 2021, doi:10.3390/v13091732_

Round 1

Reviewer 1 Report

The authors present an interesting paper on characterization od bacteriophage-host interactions on the model of Bordetella bronchiseptica bacterium and 5 phages classified as Siphoviridae. One temperate phage was described in detail, indicating that although high infectivity, lysogenic conversion is possible, potentially enhancing virulence of the bacterial host. Thus, the authors conclude that such temperate phages are not useful in phage therapy.

Generally, this is an interesting and important paper. However, major revision is required before its publication.

Specific points:

  1. Since investigated phages have been discovered and described previously [ref. 23], more information about them is required in Introduction.
  2. It is not clear why some of the investigated 5 phages were tested in some assays and other phages in different assays (for example, LK3, CN1 and FP in Fig. 1, and CN2, LK3 and MW2 in Fig. 7). All phages should be tested in all assays, otherwise the work is incomplete. If there are any special reasons why the authors investigated only some phages in specific assays, they must be clearly explained, but it is unlikely that some of phage could not be tested in basic assays.
  3. I understand that LK3 was chosen as a selected phage for more detailed analysis, however, since sequences of genomes of all 5 phages were determined, at least maps of all phages should be demonstrated, analogously as it is presented in Fig. 2 for LK3. Other maps can be shown in Supplementary materials if the authors consider they are not appropriate to the main manuscript.
  4. The terms "infected" and "uninfected" is confusing when describing lysogenic and non-lysogenic strains. "Infected" may also mean a bacterium after infection with the phage and during its lytic development. Therefore, when describing effects of prophages on host cells, I strongly recommend to use terms "lysogenic" and "non-lysogenic" rather than "infected" and "uninfected" throughout the text.
  5. In the Abstract, all 5 investigated phages should be mentioned and shortly described.
  6. Section 2.1. - describe BbChiot and Bbr3416 strain in more detail or provide relevant references if they were characterized previously.
  7. Section 2.2. - List all 5 investigated phages, and provide appropriate reference (perhaps ref. [23]).
  8. Line 155 - use appropriate subscripts when presenting chemical formulas.
  9. Several references are cited by names and years. However, in this journal, references are cited by numbers, and without such numbers, it is not possible to easily find references cited by authors and years. Please, provide appropriate numbers (lines: 232, 247, 406).
  10. Please, be sure to present names of bacterial species and genera in italic font.
  11. Line 301 - "encoding genes" is a jargon. Genes cannot be encoded, as genes code for proteins or RNAs. Therefore, replace such a statement with "genes encoding DNA polymerase, primase....." or "genes coding for DNA polymerase, primase.....". The same applies to line 424.
  12. Line 309 - Replace "LK" with "LK3".
  13. Results presented in Fig. 7 should be presented at the beginning of the Results section, just before or just after the results currently shown in Fig. 1. Again, all 5 phages should be tested in assays presented in current Figs. 1 and 7.
  14. Since a potential utility/uselessness of temperate bacteriophages in phage therapy is described by the authors, they might wish to discuss a recent paper focused on effects of temperate bacteriophages on mammalian organisms (see PMID: 34071422, DOI: 10.3390/v13061013).

Author Response

To the Editors of Viruses

25th August 2021

Re: Are Bordetella bronchiseptica siphoviruses (genus Vojvodinavirus) appropriate for phage therapy - bacterial allies or foes?

Dear Editors

Thank you for the opportunity to submit a revised version of our manuscript. We also thank the reviewers for their comments and suggestions and have responded to each of these below.

REVIEWER 1

The authors present an interesting paper on characterization of bacteriophage-host interactions on the model of Bordetella bronchiseptica bacterium and 5 phages classified as Siphoviridae. One temperate phage was described in detail, indicating that although high infectivity, lysogenic conversion is possible, potentially enhancing virulence of the bacterial host. Thus, the authors conclude that such temperate phages are not useful in phage therapy.

Generally, this is an interesting and important paper. However, major revision is required before its publication.

Specific points:

  1. Since investigated phages have been discovered and described previously [ref. 23], more information about them is required in Introduction.

Response: Thank you for the suggestion. We have added key information about examined phages described from ref. [23].

Please see lines 62-64.

  1. It is not clear why some of the investigated 5 phages were tested in some assays and other phages in different assays (for example, LK3, CN1 and FP in Fig. 1, and CN2, LK3 and MW2 in Fig. 7). All phages should be tested in all assays, otherwise the work is incomplete. If there are any special reasons why the authors investigated only some phages in specific assays, they must be clearly explained, but it is unlikely that some of phage could not be tested in basic assays.

Response: Thank you for your comment. Our previous study [23] showed that all examined B. bronchiseptica siphoviruses were highly related, showing similar morphology, but with certain variation in host range, the efficacy of plating and RFLP patterns. In addition, whole-genome sequencing of five phages reported here (LK3, CN1, CN2, MW2 and FP1) have been assessed by the International Committee for Taxonomy of Viruses (ICTV) - taxonomic proposal (2019.094B) available at:

https://talk.ictvonline.org/taxonomy/p/taxonomy-history?taxnode_id=202008066.

This proposal includes the nomination of a new genus – Vojvodinavirus comprising four new species (CN1, CN2, FP1 and MW2, each of the proposed species differs from the others with more than 5% at the DNA level as confirmed with the BLASTN algorithm). Due to high DNA sequence relatedness between phages, we decided to test only a few phages based on their subtle difference at the genome level, RFLP profiles and host ranges. We have included relevant ICTV references and taxonomic proposals from March 2020 [24, 25] which supports the rationale behind the experimental approach used in this study (please see lines 129-132 and 362-368).

  1. I understand that LK3 was chosen as a selected phage for more detailed analysis, however, since sequences of genomes of all 5 phages were determined, at least maps of all phages should be demonstrated, analogously as it is presented in Fig. 2 for LK3. Other maps can be shown in Supplementary materials if the authors consider they are not appropriate to the main manuscript.

Response: Thank you for this recommendation. We have decided not to show genome maps of other phages as they are highly similar to each other (>88.3%) and we believe these would cause unnecessary repetitiveness in the manuscript. However, to address your comment, we have included an ICTV taxonomic proposal [25] that includes the phylogenetic tree of Vojvodinaviruses and related Yuavirus phages. We believe that the phylogenetic tree better represents whole-genome sequencing results of B. bronchispetica siphoviruses and genetic relatedness between tested phages.

  1. The terms "infected" and "uninfected" is confusing when describing lysogenic and non-lysogenic strains. "Infected" may also mean a bacterium after infection with the phage and during its lytic development. Therefore, when describing the effects of prophages on host cells, I strongly recommend to use terms "lysogenic" and "non-lysogenic" rather than "infected" and "uninfected" throughout the text.

Response: Thank you for suggestion. We corrected the manuscript accordingly.

  1. In the Abstract, all 5 investigated phages should be mentioned and shortly described.

Response: Thank you. Noted and amended (please see lines 15-17).

  1. Section 2.1. - describe BbChiot and Bbr3416 strain in more detail or provide relevant references if they were characterized previously.

Response: We have added information about strains into manuscript, as advised; Please see lines 74-75 and references [26] and [27].

  1. Section 2.2. - List all 5 investigated phages, and provide appropriate reference (perhaps ref. [23]).

Response: Amended. Please see lines 82-84.

  1. Line 155 - use appropriate subscripts when presenting chemical formulas.

Response: Thank you. We have amended chemical formulas accordingly throughout the text.  

  1. Several references are cited by names and years. However, in this journal, references are cited by numbers, and without such numbers, it is not possible to easily find references cited by authors and years. Please, provide appropriate numbers (lines: 232, 247, 406).

Response: Thank you for noticing this. We have added references, as advised.

  1. Please, be sure to present names of bacterial species and genera in italic font.

Response: Thank you. We screened the entire manuscript and made sure that all bacterial species/genera are italicized. An exception are phage strains names, in which bacterial genus has to be written in regular font, according to ICTV.

  1. Line 301 - "encoding genes" is a jargon. Genes cannot be encoded, as genes code for proteins or RNAs. Therefore, replace such a statement with "genes encoding DNA polymerase, primase....." or "genes coding for DNA polymerase, primase.....". The same applies to line 424.

Response: Thank you very much for this comment. We completely agree, thus we have corrected it. This is two sentences, as suggested (please see 355 and 517 lines).

  1. Line 309 - Replace "LK" with "LK3"

Response: Thank you, we have replaced LK with LK3.

  1. Results presented in Fig. 7 should be presented at the beginning of the Results section, just before or just after the results currently shown in Fig. 1. Again, all 5 phages should be tested in assays presented in current Figs. 1 and 7.

Response: Thank you for this suggestion, we have presented results of phage sensitivity to environmental factors at the beginning of the Results section, next to whole-genome sequencing results.

  1. Since a potential utility/uselessness of temperate bacteriophages in phage therapy is described by the authors, they might wish to discuss a recent paper focused on effects of temperate bacteriophages on mammalian organisms (see PMID: 34071422, DOI: 10.3390/v13061013).

Response: Thank you for this guidance. We have briefly discussed the effects of temperate phages on the mammalian immune system. Please see lines 579-583.

Reviewer 2 Report

Aleksandra Petrovic Fabijan et al., describes Bordetella bronchiseptica specific bacteriophages and their anti- B. bronchiseptica activity including inhibition of bacterial growth, biofilm formation and reduction of already formed biofilm. Complete genome of Bordetella phage LK3 has been reported that showed integrase and repressor protein sequence presence, what indicated phage potential to lysogenize bacteria.

The manuscript is well written, but I have few specific comments:

First, the author sequenced genomes of five bacteriophages, not only LK3 and this information should be placed in the Abstract. Moreover, part of the genome of phage vB_BbrS_CN1 was already sequenced in Petrovic et al., 2017 (DOI 10.1007/s00248-016-0847-0) and this information also needs to be placed in the manuscript.

In the Materials and Methods part there is only information about bacterial strains used in the study, the source of bacteriophages is omitted and should be indicated.

The last comment is that already in Petrovic et al., 2017 (the publication of the same research group) there is an information that the CN1 phage “…is probably not useful for application as antimicrobial agent…” because the phage is able to form stable lysogens.

Similar conclusion about LK3 bacteriophage is placed in the Discussion in the present manuscript. L466-468.  In publication from 2017 the authors suggest to study endolysins of isolated bacteriophages, so my question is: Why the group characterized in detail phages, and their endolysins remained neglected?

minor comments:

All bacterial names such as Bordetella, Pseudomonas etc. need to be given in italics e.g. Bordetella phage, L21; Pseudomonas phage Yua-related bacteriophages, L493

Full names of bacterial strains should be indicated, like B. bronchiseptica ATCC 10580, L76 – not only ATCC numbers

B. bronchiseptica please write in capital letters, L176

after incubation bacteria were suspended in 0.9% (sodium-chloride?), L249

was not significantly changed, L353

is involved in a global …, L431

should be: reference strain B. bronchiseptica ATCC 10580, L454

Author Response

REVIEWER 2

Aleksandra Petrovic Fabijan et al., describes Bordetella bronchiseptica specific bacteriophages and their anti- B. bronchiseptica activity including inhibition of bacterial growth, biofilm formation and reduction of already formed biofilm. Complete genome of Bordetella phage LK3 has been reported that showed integrase and repressor protein sequence presence, what indicated phage potential to lysogenize bacteria.

The manuscript is well written, but I have few specific comments:

  1. First, the author sequenced genomes of five bacteriophages, not only LK3 and this information should be placed in the Abstract. Moreover, part of the genome of phage vB_BbrS_CN1 was already sequenced in Petrovic et al., 2017 (DOI 10.1007/s00248-016-0847-0) and this information also needs to be placed in the manuscript.

Response: Thank you for this comment. We have amended the manuscript accordingly. Please see lines 15-17 and 62-64.

  1. In the Materials and Methods part there is only information about bacterial strains used in the study, the source of bacteriophages is omitted and should be indicated.

Response: Thank you. We have added information about phages. Please see lines 82-84.

  1. The last comment is that already in Petrovic et al., 2017 (the publication of the same research group) there is an information that the CN1 phage “…is probably not useful for application as antimicrobial agent…” because the phage is able to form stable lysogens.

Response: Thank you for the comment. The comment in reference from 2017, that CN1 phage is not useful for therapy, is not based on lysogens examination, but from the similarity with Pseudomonas phage Yua, that possesses integrase. We mentioned the phage possible involvement in bacterial virulence, which was examined in the present manuscript.

  1. Similar conclusion about LK3 bacteriophage is placed in the Discussion in the present manuscript. L466-468. In publication from 2017 the authors suggest to study endolysins of isolated bacteriophages, so my question is: Why the group characterized in detail phages, and their endolysins remained neglected?

Response: We examined whether the phages are temperate since in the reference from 2017 we indicated that CN1 “is probably not useful for application”; so we wanted to confirm this assumption and to further examine phages possible involvement in the lysogenic conversion of B. bronchiseptica. The endolysins will be examined in the future study.

Minor comments:

  1. All bacterial names such as Bordetella, Pseudomonas etc. need to be given in italics e.g. Bordetella phage, L21; Pseudomonas phage Yua-related bacteriophages, L493
  2. Full names of bacterial strains should be indicated, like B. bronchiseptica ATCC 10580, L76 – not only ATCC numbers
  3. bronchiseptica please write in capital letters, L176
  4. after incubation bacteria were suspended in 0.9% (sodium-chloride?), L249
  5. was not significantly changed, L353
  6. is involved in a global …, L431
  7. should be: reference strain B. bronchiseptica ATCC 10580, L454

Response: Thank you very much for these comments, we appreciate the effort. We have made all necessary corrections proposed as “minor comments”. We have also screened the entire manuscript and made sure that all bacterial species/genera are italicized. An exception arephage strains names, in which bacterial genus has to be written in regular font, according to ICTV.

Reviewer 3 Report

Since the authors describe a novel bacteriophage, the title should be modified to indicate this.

In section 2.4, the stain should be identified.

The references should be completely overhauled to comply with the requirements of the journal. The article titles should be in lower case except the first letter and proper names.

Section subheadings should be all lower case (except the firs letter and proper names.)

The phage inhibited bacterial growth; but what was its effect on bacterial viability?

Although understandable, the English should be improved. Here are a few examples:

humans' pathogens

fragments separation

1% of agarose gel

biofilm was fixated

percentage (%) [not necessary]

Author Response

REVIEWER 3

  1. Since the authors describe a novel bacteriophage, the title should be modified to indicate this.

Response: Thank you for the comment. The phages have been partially described in the reference from 2017 [23], so we would like to keep the current title, with the indicated name of a new genus.

  1. In section 2.4, the stain should be identified.

Response: Thank you for this comment. We have used a solution of 0.4% crystal violet. A detailed description of the method used to assess the phage effect on bacterial biofilm can be found in reference [33].

  1. The references should be completely overhauled to comply with the requirements of the journal. The article titles should be in lower case except for the first letter and proper names.

Response: Thank you. We have amended the references throughout the manuscript to comply with the requirements of the Viruses journal. The title is written in lower case except for the first letter and bacterial name.

  1. Section subheadings should be all lower case (except the firs letter and proper names.)

Response: Thank you, we have checked all subheadings and made sure they are all written in lower case (except first letter and proper names).

  1. The phage inhibited bacterial growth; but what was its effect on bacterial viability?

Response: Thank you for this comment. We have only examined inhibition of bacteria growth and biofilm, therefore, we do not know what is the phage effect on bacterial viability. Due to phage temperate nature, it would be very interesting to examine it in future studies.

  1. Although understandable, the English should be improved. Here are a few examples:

humans' pathogens

fragments separation

1% of agarose gel

biofilm was fixated

percentage (%) [not necessary]

Response: Thank you very much for these comments, we appreciate the effort. We have made all necessary corrections.

Reviewer 4 Report

The authors characterize several phages specific against Bordetella bronchiseptica. They show that the phages are temperate and can modulate the properties of their host by lysogenic conversion. The scientific experiments are well designed, well executed and the experimental results are solid, facts that should lead to a very strong manuscript that should be easy to publish.

Unfortunately, overall, the paper is very poorly written. The authors seem to lack the necessary scientific vocabulary to concisely describe their results. This is especially true in the discussion and conclusion section of the manuscript. And there is more to it than just a struggle with the English language. That all six authors, who according to the “authors contributions” were involved in reviewing and editing the final manuscript, managed to overlook that the name of their host bacterium was misspelled on five occasions, shows a certain carelessness in the preparation of the manuscript.

Major points:

The term „phage-infected bacteria” is not an appropriate term as it can be misleading and is wrong in the context the authors use it. The correct term would be “lysogen”. When a phage like LK3 integrates into the genome of its host, all progeny of that bacterium are LK3 lysogens. In all the experiments were the authors address the effects of the lysogenic conversion, they compare a lysogen with a non-lysogen (in the words of the authors “phage-infected bacteria” with “uninfected bacteria”). This should be corrected throughout the manuscript.

Line 288, “of double-stranded linear DNA with”? What does the term linear mean in this context? Linear DNA as opposed to circular DNA? The phage DNA is always linear when packaged into the phage head, but most likely circular when present within the bacterial cell after infection and then linear again when integrated into the chromosome of the host. The term “linear” should be omitted from the text in this context. Especially when the authors then represent the DNA as a circle and show that the DNA in the phage head is circularly permuted and in later experiments show that they detect “circular plasmid DNA” within the host cells.

Lines 468 to 469. “In addition, pac site was previously confirmed [23], so the phages are able for horizontal gene transfer, i.e. for both generalized and specialized transduction.” This statement in the discussion is scientifically false. Spezialized transduction is in no way linked to the presence of a pac site (see phage lambda as the most prominent example). Also, while the presence of a pac site indicates a headful packaging mechanism, a direct link to generalized transduction is also not correct (see wt T4 as a good example).

Minor corrections:

Line 24, either “LK3 phage forms” or “LK3 phages form”

Line 33, “human pathogens”, (not human´s)

Line 45, “In the last few years”

Line 51, “phages from the Podoviridae family”

Line 79, and later in the text „CsCl2

Line 85, “McLaughlin (2007)” typo in the name

Lines 85 to 88, the sentence is convoluted and should be rewritten. I propose:

In sterile microtiter 96-well flat bottom plates, 50 µl of double strength LB were inoculated with a final bacterial concentration of ~ 5 x 105 CFU ml-1 and an equal volume of ten-fold serial phage dilutions in SM buffer were added (final volume 100 µl).

Line 86, and later in the text, “double strength LB”, not “strengthen”

Line 145, “destained”

Line 156, “that did not infect bacteria”, “didn’t” is considered slang and should not be used in a scientific context.

Line 170, „enzyme does not“, the same goes for “doesn’t”

Line 176, “B. bronchiseptica”, the B. should be capital

Lines 196 to 197, “Experiment was carried out in triplicate and three independent experiments.”? The sentence does not make sense in this form and should be rephrased.

Line 213, “Susceptibility of phage infected bacteria to antibiotics”

Line 226, “B. bronchiseptica” typo in the name

Line 247, “by Jain i Chen (2007)” ? i = and?

Lines 265 to 266, „The results were presented as“

In the legend to Figure 1, B. bronchiseptica should be set italic.

Line 292, “genome also contains eight ORFs”

Line 317 to 318, (ATCC 10580+, Bbchiot+ i Bbr3416+) ? i = and?

Line 328, “Bbchiot i Bbr3416 showed” as above?

Legend to Figure 4, “Quality” typo

Line 333, “various starting bacterial numbers”

Line 341, “It is also interesting to notice” it’s is also considered slang

Lines 352 to 353, “bacteria remained sensitive“

Line 364, “urea reduced the number of FP1”

Lines 386 to 387, “different than in planktonic cells [36].”

Line 390, “LK3 phage also uses these surface structures in the process of adhesion to the host”

Line 393, “reduced, as temperate phages rather integrate”

Lines 432 to 433, “This enzyme was previously detected”

Line 455, “phage did not show equal” as above

Line 475, “B. bronchiseptica” typo in the name

Line 491, “activity of B. bronchiseptica” typo in the name

Line 492, “B. bronchiseptica specific” typo in the name

Author Response

REVIEWER 4

The authors characterize several phages specific against Bordetella bronchiseptica. They show that the phages are temperate and can modulate the properties of their host by lysogenic conversion. The scientific experiments are well designed, well executed and the experimental results are solid, facts that should lead to a very strong manuscript that should be easy to publish.

Unfortunately, overall, the paper is very poorly written. The authors seem to lack the necessary scientific vocabulary to concisely describe their results. This is especially true in the discussion and conclusion section of the manuscript. And there is more to it than just a struggle with the English language. That all six authors, who according to the “authors contributions” were involved in reviewing and editing the final manuscript, managed to overlook that the name of their host bacterium was misspelled on five occasions, shows a certain carelessness in the preparation of the manuscript.

Major points:

  1. The term „phage-infected bacteria” is not an appropriate term as it can be misleading and is wrong in the context the authors use it. The correct term would be “lysogen”. When a phage like LK3 integrates into the genome of its host, all progeny of that bacterium is LK3 lysogens. In all the experiments were the authors address the effects of the lysogenic conversion, they compare a lysogen with a non-lysogen (in the words of the authors “phage-infected bacteria” with “uninfected bacteria”). This should be corrected throughout the manuscript.

Response: Thank you for your suggestion. We have corrected this - instead of infected strain we used ‘lysogenic’ and instead of non-infected strain, we used ‘non-lysogenic.

  1. Line 288, “of double-stranded linear DNA with”? What does the term linear mean in this context? Linear DNA as opposed to circular DNA? The phage DNA is always linear when packaged into the phage head, but most likely circular when present within the bacterial cell after infection and then linear again when integrated into the chromosome of the host. The term “linear” should be omitted from the text in this context. Especially when the authors then represent the DNA as a circle and show that the DNA in the phage head is circularly permuted and in later experiments show that they detect “circular plasmid DNA” within the host cells.

Response: Thank you for this suggestion. We have omitted “linear” from the text to avoid confusion.

  1. Lines 468 to 469. “In addition, pac site was previously confirmed [23], so the phages are able for horizontal gene transfer, i.e. for both generalized and specialized transduction.” This statement in the discussion is scientifically false. Spezialized transduction is in no way linked to the presence of a pac site (see phage lambda as the most prominent example). Also, while the presence of a pac site indicates a headful packaging mechanism, a direct link to generalized transduction is also not correct (see wt T4 as a good example).

Response: We agree that this sentence is clumsily formulated – headful packaging in phages is not directly linked to specialised transduction. However, phages that possess integrase are known to be prone to specialized transduction, including lambda phage (DOI: 10.1126/science.aat5867). The presence of pac site and thus possible recognition of pseudo-pac sites (pac site homologs) in bacterial DNA by terminase indicate a capacity for phage-mediated generalized transduction. To avoid any further confusion, we have rewritten this sentence (please see 517-518 lines).

Minor corrections:

  1. Line 24, either “LK3 phage forms” or “LK3 phages form”
  2. Line 33, “human pathogens”, (not human´s)
  3. Line 45, “In the last few years”
  4. Line 51, “phages from the Podoviridae family”
  5. Line 79, and later in the text „CsCl2“
  6. Line 85, “McLaughlin (2007)” typo in the name
  7. Lines 85 to 88, the sentence is convoluted and should be rewritten. I propose:

In sterile microtiter 96-well flat bottom plates, 50 µl of double strength LB were inoculated with a final bacterial concentration of ~ 5 x 105 CFU ml-1 and an equal volume of ten-fold serial phage dilutions in SM buffer were added (final volume 100 µl).

  1. Line 86, and later in the text, “double strength LB”, not “strengthen
  2. Line 145, “destained
  3. Line 156, “that did not infect bacteria”, “didn’t” is considered slang and should not be used in a scientific context.
  4. Line 170, „enzyme does not“, the same goes for “doesn’t
  5. Line 176, “B. bronchiseptica”, the B. should be capital
  6. Lines 196 to 197, “Experiment was carried out in triplicate and three independent experiments.”? The sentence does not make sense in this form and should be rephrased.
  7. Line 213, “Susceptibility of phage infected bacteria to antibiotics”
  8. Line 226, “B. bronchiseptica” typo in the name
  9. Line 247, “by Jain i Chen (2007)” ? i = and?
  10. Lines 265 to 266, „The results were presented as
  11. In the legend to Figure 1, B. bronchiseptica should be set italic
  12. Line 292, “genome also contains eight ORFs”
  13. Line 317 to 318, (ATCC 10580+, Bbchiot+ i Bbr3416+) ? i = and?
  14. Line 328, “Bbchiot i Bbr3416 showed” as above?
  15. Legend to Figure 4, “Quality” typo
  16. Line 333, “various starting bacterial numbers”
  17. Line 341, “It is also interesting to notice” it’s is also considered slang
  18. Lines 352 to 353, “bacteria remained sensitive“
  19. Line 364, “urea reduced the number of FP1”
  20. Lines 386 to 387, “different than in planktonic cells [36].”
  21. Line 390, “LK3 phage also uses these surface structures in the process of adhesion to the host”
  22. Line 393, “reduced, as temperate phages rather integrate”
  23. Lines 432 to 433, “This enzyme was previously detected”
  24. Line 455, “phage did not show equal” as abov
  25. Line 475, “B. bronchiseptica” typo in the nam
  26. Line 491, “activity of B. bronchiseptica” typo in the name
  27. Line 492, “B. bronchiseptica specific” typo in the name

Response: Thank you very much for these comments, we appreciate the effort. We have made all necessary corrections.

Round 2

Reviewer 1 Report

I am satisfied with authors' responses and modifications introduced into the manuscript. I have no further comments, and I recommend publication of the revised paper.